# A Proof-of-Concept Pilot for an Intervention with Pregnant Mothers Who Have Had Children Removed by the State: The 'Early Family Drug and Alcohol Court Model'

**Mike Shaw**

Tavistock & Portman NHS Foundation Trust, London NW3 5BA, UK; mshaw@tavi-port.nhs.uk

**Abstract:** This paper describes a 'proof-of-concept' pilot of the 'Early FDAC model'. The evaluated Family Drug and Alcohol Court (FDAC) model, on which Early FDAC is based, is summarised and the rationale for introducing the pilot variation is set out. This short paper describes the learning from the pilot that set out to work with 30 families across three FDAC teams between 2015 and 2019. At the time of the pilot, there were, and remain, few other interventions in England for pregnant mothers who have already had children removed. An adaptation of the evaluated FDAC model suggested itself because of the overlap with families in public law care proceedings and emerging evidence that FDAC delivers a better experience of justice for families and professionals, better outcomes for children and families and better use of public money. Pilot families were engaged as soon as possible in the pregnancy (hence 'Early'), and continued to receive support for up to two years. The process started in pre-proceedings with the aim of avoiding court. Where proceedings were necessary, cases were heard in an FDAC court, with provision for a post-proceedings phase. There were problems with recruitment and engagement and families had fewer 'solvable problems'. Nevertheless, outcomes were promising, with 18 families keeping their children. This represents one-third of the referred families and almost two-thirds of the families who undertook a 'Trial for Change'.

**Keywords:** pilot; pregnant mothers; children removed by the state; family drug and alcohol court (FDAC); Trial for Change; recruitment and engagement pathways; promising outcomes





## 1. Introduction

The Family Drug and Alcohol Court (FDAC) brings a problem-solving [1] approach to care proceedings, providing families with the chance to work intensively with a specially-trained judge and therapeutic team. FDAC works on the assumption that parents want to meet their children's needs, but are prevented from doing so by problems with intimate partner abuse and/or substance misuse and/or mental illness, all of which are often further complicated by poverty. The FDAC model provides parents with the best possible chance to solve their problems while testing whether they can meet their children's needs in a timescale compatible with those needs. Adapted from similar courts in the US [2], FDAC first opened in London in January 2008, and there are now 9 FDAC teams serving 13 courts and 20 local authorities across England and a further team and court in Northern Ireland.

'Early FDAC' is a modified version of the evaluated FDAC model, which aims to reduce repeated care proceedings and removals for mothers who have already had children removed by the state, and who continue to have problems that are likely to interfere with their parental capacity, and who become pregnant and, thereby, are at risk of losing further children.

In the Early FDAC model, mothers (and their partners) begin work with the therapeutic team as early as possible in the pregnancy, and continue to receive support for up to two years. The process starts in pre-proceedings, and if care proceedings are necessary, cases are heard in an FDAC court and there is provision for a post-proceedings phase.

This short paper describes the learning from a 'proof-of-concept' pilot of the Early FDAC model that set out to work with 30 families across three FDAC teams based in London (pilot 1), the Southeast (pilot 2) and the Midlands (pilot 3) and was completed over the period 2015–2019.

## 2. The FDAC Model

### 2.1. Overview

The evaluated FDAC model has been described as "arguably the most radical development in family justice since the Children Act 1989" [3].

To anyone familiar with standard public law care proceedings, one of the most striking differences about FDAC is the atmosphere in the courtroom (see evaluation below). The judge converses with the parents directly rather than through their representatives, and wherever possible, the truth is uncovered through collaboration rather than contested position taking. Parents are treated with compassion and respect and are encouraged to believe that their family's prospects can be improved if they are prepared to be open and honest and work hard.

Participating local authorities select cases for FDAC where concerns about parental substance misuse have contributed to proceedings being issued. Parents have to 'opt in' during the initial court hearing and, following an assessment by the therapeutic team, 'sign up' at a further initial hearing a few weeks later. Parents can decide to revert to standard care proceedings at any point.

The FDAC therapeutic team is composed of children's social workers and substance misuse specialists, with some input from 'parent mentors' (parents with their own experience of substance misuse and/or care proceedings), an adult psychiatrist and a child and adolescent psychiatrist or psychologist.

Between the two initial hearings, the therapeutic team reads the background papers and carries out a very thorough whole-day assessment of the parents, during which parents are once more encouraged to be open, honest and take some responsibility for the problems that have brought them into proceedings.

In consultation with the parents, the local authority, the children's guardian and other agencies, the team designs an individualised 'Trial for Change' that will: give parents the best possible chance to overcome their problems; test whether they can overcome their problems and meet their children's needs in a timescale compatible with their children's needs; and make use of resources that can be accessed in a timely fashion from a network of partner agencies or the FDAC team itself.

The Trial for Change requires parents and professionals to work towards shared goals and timescales, but asks parents to take a lead in establishing these and asks professionals to support them in meeting these. Once there is agreement from all parties, the parents sign up at the second initial hearing and the Trial for Change thereafter takes on the authority of the court. There are four overlapping elements to the parents' work: (i) desisting from damaging behaviours (such as substance misuse, or intimate partner abuse); (ii) addressing the trauma that was contributing to those behaviours (which often include the parents' own unresolved histories of childhood abuse and neglect, and/or later losses and abuses); (iii) strengthening relationships (between parents and children, and between parents and their extended family); and (iv) building a child-centred lifestyle (including engaging with education, training and employment).

Much is required of parents engaging in FDAC. They are expected to keep a diary and make time for several contacts a week with their children, fortnightly court attendance, three monthly hair-strand drug and alcohol tests, monthly alcohol blood tests, weekly or twice weekly body fluid drug tests and breath alcohol tests, weekly therapeutic meetings with their FDAC keyworker, and at least one other weekly therapeutic meeting, usually with the local substance misuse treatment agency.

Most parents go on to engage with some combination of Alcoholics Anonymous, anxiety groups, Circle of Security [4], Video Interaction Guidance [5] and/or Social Behavioural

Network Therapy [6]. Some parents go on to engage with several months of intensive and demanding treatment in a residential, or more often community setting [7].

Unlike standard public law care proceedings, the FDAC model allows for a close working relationship between the therapeutic team and the judge. In addition to providing reports for the court (shared with all the parties), the therapeutic team briefs the judge in chambers before every hearing. Furthermore, the team is always present in court and available to meet with the parties outside of court.

Families work with the same judge throughout the proceedings, and in addition to the standard hearings with lawyers, there are fortnightly hearings without lawyers. These non-lawyer hearings last approximately half an hour and are attended by the parents, the parents' FDAC keyworkers and the children's social worker and guardian. Parents are encouraged to tell the judge what is going well and/or not going well with the Trial for Change, and what they are hoping to achieve in the next two weeks. The judge will encourage the parents, but also remind them of the children's and court's timescales and how the court's decision depends on the parents evidencing they can change. Bullet points from the hearings are distributed to the parties so there should be no 'surprises' about where things stand.

Most proceedings begin with children being removed from their parents' care and it is usually not until week 18 in the proceedings that the therapeutic team will give an indication whether reunion will be possible. A decision to permanently remove children is normally made within the 26 week time limit, but where children are returning home there is normally a short extension to support and test out the process. Most proceedings end by consent and prolonged contested hearings are very rare.

### 2.2. Learning from Evaluations of FDAC

FDAC has a growing evidence base [8] supporting a view that FDAC delivers a better experience of justice for families and professionals [9–11], better outcomes for children and families [10,12] and better use of public money [13].

Harwin et al. 2014 [10] conducted structured observations of London FDAC hearings (114 cases) and found that more often than not judges talked to parents directly, invited parents' views, expressed interest in progress, praised parents, urged parents to take responsibility and used a problem-solving approach (i.e., used court time to try to resolve problems). Tunnard et al. 2016 [11] used a similar structured observational method across 10 different FDAC courts (40 cases) with very similar results. While these findings seem to confirm that FDAC is operating in keeping with its principals, cases were a "cross-section" rather than randomly selected and there was no comparison with judicial behaviour in standard proceedings.

Harwin et al. 2011 and 2014 [9,10] reported on semi-structured interviews with 52 parents. They found that parents spoke warmly about the judges, describing them as 'reasonable', 'encouraging', 'sensitive' and 'calm'. They said the judges 'treated you like a human being', 'talked about normal things' and 'put you at your ease'.

Earning the praise of the judge tended to be valued much more than praise from other professionals.

> *"No-one praised me before. My solicitor does, but I expect that. When I go to court I come out feeling really happy. My social worker never praises me, or never says it in a way that feels nice."*

Parents also appreciated judges being firm but fair.

> *"At first I didn't like him because he was honest. He was saying it how it was and it was bad. It was horrible. But now I know it was the truth."*

Two-thirds of the parents interviewed were positive about the non-lawyer review hearings. They liked their informality and the fact that they 'stopped problems from escalating' and 'kept everybody up to date'.

Some parents had prior experience of standard proceedings and 15 mothers had previously had children removed by the court. These parents reported that in standard proceedings, they had seen the judge only rarely, had felt treated as 'junkies' or 'prostitutes', and were made to feel that there was very little chance of being allowed to keep their child. They also talked of feeling unsupported.

> *"I've been to an ordinary care case before and normally you wouldn't get any advice. This is what I think I need. In the other court no-one actually works with you. All that the social workers said was 'go to rehab'."*

The parents were "overwhelmingly positive" in their comments about the FDAC therapeutic team. Parents liked 'being talked to as normal' and 'not being judged straight away'. FDAC 'listened' and 'were always explaining things'. 'Honest', 'strict', 'supportive' and 'kind' were the words used most often to describe individual team members. Parents said that the honest way in which team members spoke to them was particularly helpful in enabling them to talk about their problems in an open and realistic fashion.

> *"Instead of fibbing we're encouraged to be honest and if we relapse, or lapse even, we're told it wouldn't be the end of it, because they would work with us about that. They were being honest with us and making it easier for us to be honest with them."*

Parent valued the parents mentors as someone to identify with.

> *"What's good about it is hearing someone else's experience and how they came through it. FDAC are all professionals, but the mentor is just like me. It helped a lot".*

While it was rare for parents to criticise the way they were treated, it is important to know that the researchers failed in their efforts to interview parents who had opted or dropped out of FDAC.

Finally, Harwin et al. 2011 and 2014 [9,10] carried out semi-structured interviews and focus groups with a large group of professionals (140 individuals), including FDAC judges and court officials, FDAC team members, children's guardians, local authority social workers and managers, local authority and family lawyers, substance misuse treatment services and commissioners.

Professionals consistently commented that FDAC hearings were less adversarial than ordinary care proceedings.

> *"In ordinary proceedings it is very much 'us and them'. It is very good for parents to see the lack of antagonism between the professionals in these cases."* (family lawyer)

On the other hand, lawyers said that they are not prevented from advocating on behalf of their clients, or from raising issues of concern or contesting matters that need to be challenged.

Initial concerns about the judges speaking directly to parents without their legal representatives present (in the non-lawyer hearings) receded as familiarity with the model increased. This was especially the case as it became clear that direct discussion with the judge boosted parents' confidence and encouraged them to take more responsibility for their behaviour.

> *"I have never heard parents speak so openly in court as they do in FDAC. Their confidence develops. They move from rigidity to feeling more relaxed and they build a relationship with the Judge."* (local authority social worker)

> *"Clients in FDAC feel, not exactly relaxed, but they seem to take on board things a little bit more. They seem to understand a bit better why they are doing something and they are happier with the process, even if it is not something they want."* (adult treatment service)

Professionals valued the fact that the FDAC therapeutic team not only assessed parents but also worked directly with them, as part of a Trial for Change.

*"Their model of really intense support for parents to think about themselves and why they behave as they do is really important. For many parents it is the first experience of someone getting them to think about themselves in this way. Traditional community models of treatment don't have the time or capacity to do that sort of intensive work for three months—but that's the effort needed to help someone completely change their lifestyle."* (social work manager)

Professionals consider that the regular court monitoring of the parents' progress under the intervention plan, combined with the team's regular testing for drug and alcohol use, was a good test of the parents' capacity to change their behaviour.

*"If parents have all the services they need offered to them, but still cannot control their substance misuse, this helps them accept that they cannot care for their child."* (family lawyer)

There was "unstinted praise" from professionals about the skilful way in which FDAC liaises with the different services identified in a parent's intervention plan. The team was seen as knowledgeable, good at co-ordinating and reducing fragmented responses and duplication of effort.

*"It is so much easier when FDAC is involved—everyone is at meetings, there is a clear plan, you don't have to scrabble around for experts or argue about resources. And a small point, I know, but they make sure the appointments don't clash. This sort of joining up between services doesn't happen in other cases."* (family lawyer)

Most professionals commented favourably on the impact of the FDAC team's work on their workload. The general view of guardians, lawyers for parents and children, and adult treatment providers is that there is less work for them in FDAC cases, largely on account of FDAC taking the lead in co-ordinating activity around the case. The views of social workers were more mixed—while they valued the regular contact with the team, they struggled to find time for the fortnightly non-lawyer hearings.

While there were very few criticisms from professionals, it is likely that the generally favourable reception of the model made professional less willing to talk about their doubts.

Evaluations of FDAC have also addressed the issue of whether this approach delivers better outcomes for children and families. Harwin et al. 2011, 2014, and 2016 all report on different stages of a case comparison cohort study conducted over an 8 year period. By the time of the 2016 study.

The FDAC cohort was a consecutive series of the first 140 families referred to the London service in the period 2008–2012 (90 who were in the earlier studies, plus 50 more), while the comparison cohort was 100 substance-misusing families from 3 neighbouring local authorities where standard care proceedings had been issued during the same period.

Analysis showed that apart from ethnicity (the FDAC cohort had more White mothers and children), the two cohorts were similar in their sociodemographic profiles and psychosocial difficulties.

By the end of proceedings, a significantly higher proportion of FDAC than comparison families were reunited or continued to live together (37% vs. 25%).

For this final stage of the study, researchers followed mothers and children post-proceedings, using data from the local authority files, cross-referenced with the CAFCASS database (to pick up families that changed local authorities).

Mother's outcomes post-proceedings were defined as 'good' if during the 3 year follow-up period: there was no maternal substance misuse relapse; no permanent placement change for a child or children; and no return to court.

Using 'survival analysis,' they estimated that a significantly higher proportion of FDAC than comparison mothers who had been reunited with their children at the end of proceedings would experience a 'good outcome' at 3 year follow up (51% vs. 22%).

Combining these outcomes, a significantly higher proportion of the total FDAC cohort than the comparison cohort were reunited or continued to live with their children at the

end of proceedings, AND would be likely to achieve a 'good outcome' at 3 year follow up (19% vs. 5.5%) (see Figure 1).

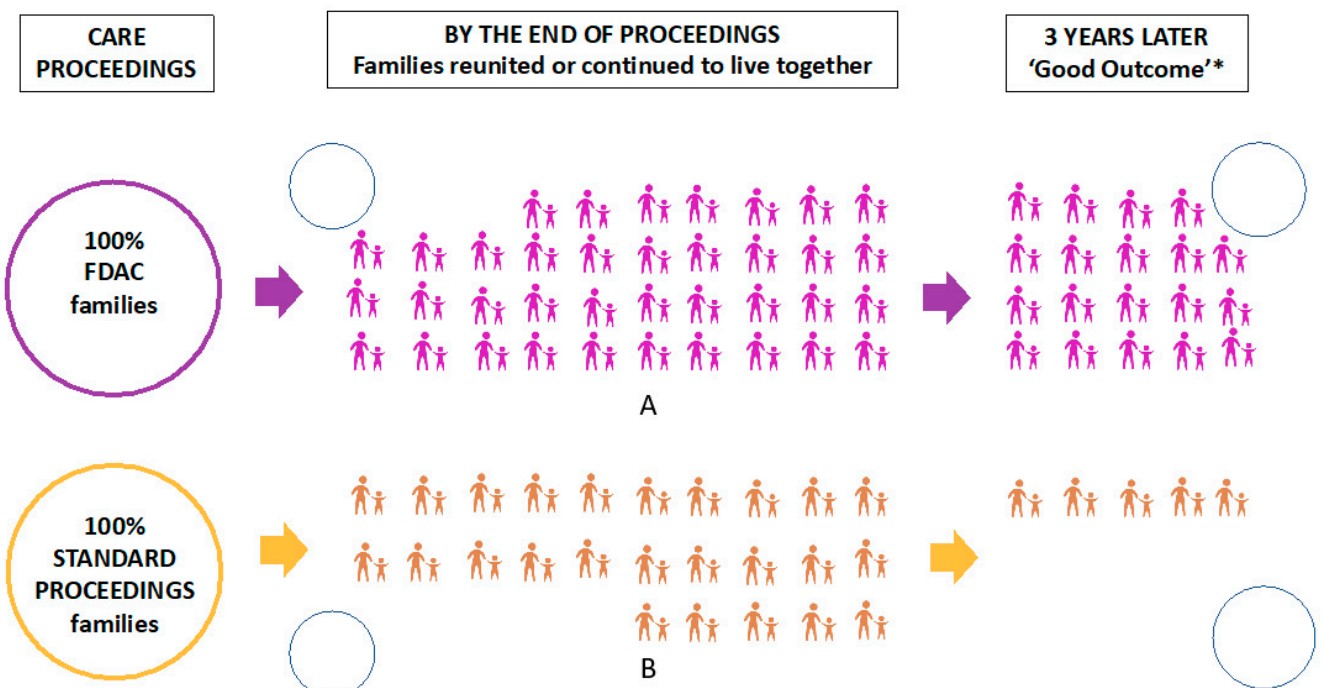

*Where 'Good Outcome' is defined as no relapse of the parent's problems, AND no change of placement, AND no return to court
Where [A] 19% Good Outcome in FDAC is calculated (51% X 37%=18.87%) & [B] 5.5% Good Outcome in standard proceedings is calculated (22% X 25%=5.5%)

**Figure 1.** Outcomes by the end of proceedings and 3 years later.

A body of largely North American research shows that, compared to conventional courts, problem-solving courts' outcomes for children and families are consistently better at the end of proceedings [14,15]. However, longer-term follow-up outcomes are less well studied and the findings are more equivocal [16]. It follows that the 3 year good outcomes demonstrated by Harwin et al. 2016 are encouraging but need replication.

Finally, there is a single economic modelling study that suggests that FDAC represents a better use of public money [13]. Reeder and Whitehead combined data from the Harwin et al. cohort studies with published economic data. Their analysis focused on the direct costs and savings to local authorities and other state stakeholders and did not include wider savings and benefits that could be attributed to societal outcomes (such as citizens' well-being). They estimated that for every £1 spent on FDAC, £1 would be saved within 2 years, and £2.30 within 5 years.

### 3. The Early FDAC Pilot

The Early FDAC pilot was devised in the context of a growing understanding of the scale of the problem with families who repeatedly have children removed by the state [17–19]. The question became what, if anything, could be done to reverse this trend.

In addition to evaluations of the FDAC model, a number of other factors influenced the development of the Early FDAC pilot. By the early 2010s, there were already a small number of services in England offering assistance to parents whose children had been removed. Some, such as Pause [20], were conditional on mothers being willing to use long-acting reversible contraception (LARC). Furthermore, while there were examples of European [21] and American [22] services working with pregnant substance-misusing mothers, there were no services in England specifically for mothers who have already had children removed, and were now pregnant and at risk of losing their unborn children.

Adapting the FDAC model had merit because of the overlap with families in care proceedings and the intervention's emerging evidence base. Furthermore, one of the

recommendations of the FDAC evaluators had been that better outcomes might be possible, if intervention started during pregnancy and continued for a year after the end of care proceedings [10].

It is also relevant that the pilot was developed in the context of reform following the Family Justice Review [23], which cut the length of care proceeding, which had previously averaged 52 weeks [23], to 26 weeks [24], and emphasised the pre-proceedings process as a means to prevent or prepare for care proceedings [25,26]. The challenge for the pilot was to give a particularly vulnerable group of parents a long-enough Trial for Change without eroding the children's or the court's timescales.

The vision was to intervene as early (hence the name 'Early FDAC') as possible during the second trimester of pregnancy and offer up to 2 years support.

If parents achieved sufficient change during the pre-proceedings element of their Trial for Change, their newborn babies would be more likely to remain in their care at birth, with Early FDAC continuing to support them on an informal basis. Whereas, if care proceedings were necessary, the case would be heard in an FDAC court and there would still be room for post-proceedings support [27].

In creating the new model, it was possible to draw on the London FDAC's experience of working in pre-proceedings [27,28] (including with pregnant mothers), and Gloucestershire FDAC's experience of providing post-proceedings support loosely based on the Nurse Family Partnership [29,30] model (known in the United Kingdom as Family Nurse Partnership).

The opportunity for funding came with the launch of the Department of Education's Innovation Fund (2015).

### 3.1. Learning from the Early FDAC Pilot: A Proof-of-Concept Review

#### 3.1.1. Evaluation

This section of this paper offers a review of the Early FDAC 'proof of concept'. It is based on outcome data provided by the services themselves and the author's contemporaneous experience as co-director of the FDAC National Unit and Clinical Lead for the Early FDAC pilot 2 team, supplemented by the reflections of practitioners spanning all three pilot site teams[1].

#### 3.1.2. Recruitment

Recruiting suitable cases for the Early FDAC pilot proved very difficult. Recruitment relied on local authorities being willing to pay part of the costs involved in opting for the FDAC route and, despite agreeing to be partners for the pilot, participating local authorities varied on whether or not they were willing or able to initiate pre-proceedings during pregnancy, with some waiting until children were born and going directly into proceedings. We were fortunate that pilot 3 retained senior-level commitment to the project and developed a robust pre-proceedings referral route. This was partly explained by the fact that FDAC had become an integral part of the local authority's children's services provision, but also by the fact that they had already established a culture of innovative work with pregnant mothers already accessing some of their, or related, services. Despite these advantages, across the project as a whole, it was rare for mothers to be referred before the third trimester of pregnancy, which meant that much valuable Early FDAC working time was lost.

#### 3.1.3. Engagement

Considerable effort was made to engage referrals including printed leaflets about the programme written in an accessible and sympathetic style. A member of the team

---

1    The Early FDAC pilot was due to include a full evaluation. However, this element of the project was unable to be completed within available time and resources. Data collected by the pilot teams is presented and synthesised here as a means of sharing insights gained from this initiative to encourage further innovation in the family justice system.

would visit mothers at home to describe the programme and answer their questions before assessment was undertaken. In all cases, there would be liaison with the referring local authority, and in instances where permission was given, other connected professionals would be contacted in an effort to facilitate engagement. Mothers were always encouraged to obtain independent legal advice. They were not always able to access this, however, because of the limited availability of legal aid payments. The impact of this was that, in contrast to main FDAC provision, local lawyers were insufficiently aware of the Early FDAC pilot. In the evaluated FDAC model, lawyers have often played a key role in strongly advising their clients to take up the FDAC offer.

In contrast to the experience with the evaluated FDAC model, many parents were less convinced about the merits of engaging with Early FDAC and there were lots of early drop outs. While participating local authorities referred 53 pregnant mothers to Early FDAC, only 36 (68%) completed an assessment and only 28 (53%) undertook a Trial for Change (see Table 1 below); by comparison, 90 out of the first 106 of cases referred to the evaluated model of FDAC (85%) undertook a Trial for Change [10].

**Table 1.** Outcomes.

|  | Pilot 1 | Pilot 2 | Pilot 3 | Totals |
| --- | --- | --- | --- | --- |
| Numbers of families referred but did not undertake assessment | 7 | 2 | 8 | 17 |
| Number of families assessed but did not adequately engage with a Trial for Change and eventual outcome unknown | 2 | 1 | 5 | 8 |
| Number of families engaged where children remained in parents' care | 3 | 3 | 12 | 18 |
| Number of families engaged where children were removed | 1 | 2 | 7 | 10 |
| Totals | 13 | 8 | 32 | 53 |

It is likely that parents who have had one or more of their children removed have a more difficult relationship with local authority and other professionals. Other research has shown that such parents often have a complex history of 'service non-engagement' [31]. However, there appeared to be a deeper problem in this case, which was that parents themselves did not take pre-proceedings as seriously as proceedings. In the evaluated FDAC model, parents face the very real threat of losing their children. The starkness of this possibility and their resultant vulnerability can often lead them to articulate a very particular desire to change. In turn, this provides the FDAC therapeutic team with a very specific opportunity to respond to that desire and to offer the support required. At the same time, all parties are made aware that the judge in the FDAC proceedings will take the view that whatever they decide to do, there will be consequences. In other words, the stakes seem higher in the evaluated FDAC model than in the Early FDAC pilot.

A more psychoanalytic way of looking at the same phenomena would be to say that parents feel 'contained' in the presence of 'the FDAC parental couple'—the curious, compassionate and encouraging maternal figure of the therapeutic team ("you are not alone we can help") working in harmony with the firm but fair paternal figure of the court ("there will be consequences"). When things go well, parents internalise and integrate this vital balance of qualities into their own parental style.

3.1.4. Different Clientele

As outlined above, the Early FDAC pilot was available to mothers who had already had children removed by the state (either by court order or by consent), and who continued to have problems that were likely to interfere with their parental capacity, and who were now pregnant and at risk of losing further children. By contrast with the evaluated FDAC

model, parenting problems linked to substance misuse were not an explicit part of the Early FDAC pilot criteria. This is likely to have had a direct impact on referral pathways into the pilot and the nature of engagement of those thus referred.

Drugs and alcohol appear in the FDAC name, and problems with drugs and alcohol are the epitome of a 'solvable problem'. There are readily available evidence-based interventions and reliable biochemical measures of change to capture these [32]. While it is true that recovery takes years, it is not unreasonable for parenting capacity to improve within months of achieving sobriety. Whilst remaining open to parents with substance misuse problems, the Early FDAC pilot sought to engage with a wider range of problems impeding parenting capacity, many of them less readily 'solvable'.

In general, the parents engaging with Early FDAC were more likely to be younger (in their 20s rather than 30s) and more emotionally vulnerable when compared to parents engaging with FDAC. Most of them had suffered abuse and neglect as children [31]. This can create an emotional shutting down or 'mindlessness' [33], where there is a turning away from reality, more wishful, even magical, thinking, and an inability to learn from trial and error. Once established, mindlessness opens the door to 'repetition' [33], where distress gets acted out through highly repetitive and destructive behaviours. The therapeutic challenge becomes how to gradually encourage a capacity to think, to help reconnect parents with their pain and introduce a realistic sense of their own destructive behaviour without creating further avoidance and/or total collapse. There is no simple solution, but it is likely that a sustained period of intervention—of up to two years—is more likely to be effective.

Further, the experience of unresolved and complex grief was universal among Early FDAC parents, with some pregnancies arising during the care proceedings in which their older children were removed. While the desire to have more children is very understandable, it represents a desire for the "triumph of hope over experience" (a comment about second marriages attributed to Samuel Johnson by James Boswell), with a remarkable number of such children named 'Hope', 'Faith', 'Chance', 'Destiny' or 'Paige' (as in 'new page').

Many Early FDAC parents experience mental health problems. Some are 'solvable' problems akin to substance misuse, given that there are readily available evidence-based interventions for anxiety, depression and less complex post-traumatic disorders, and that measurable change is achievable in months. On the other hand, while long-term effects of complex trauma are treatable, access to care is very patchy, referral and treatment pathways long, and measurable progress normally takes years [34].

Finally, intimate partner abuse may be a more readily identifiable problem than for example complex grief, but it is one that is equally hard to solve. Shame, guilt and fear mean that its effects are difficult to measure, evidence-based treatments are less well established and timelines for treating perpetrators tend to be longer than most children or family courts can wait. Overall, as the Early FDAC pilot included fewer parents with substance misuse, and more problems with complex trauma and intimate partner abuse, it was always likely that achieving change with this group would take longer and be harder to achieve and measure.

### 3.1.5. Outcomes

The limited outcome data presented below suggest that, for all its challenges, the Early FDAC pilot holds some promise as a form of innovation in family justice. The target was to work with 30 families, and by the end of the pilot, 53 families had been referred, of which 36 underwent an assessment and 28 undertook a Trial for Change (see Table 1). Significantly, 18 families went on to keep their children. This represents one-third of the 53 referred families and almost two-thirds of the 28 families who undertook a Trial for Change. These rates compare favourably with standard repeat care proceedings across the country where approximately one-sixth of families keep their children [35]. They are similar to the outcomes reported elsewhere in this issue [36] by other services working with similar

populations of mothers. However, such comparisons should be approached with caution given that it is not clear how representative the pilot families were of the national repeat proceedings population and/or comparable to the study population recently described by Cox et al. [36].

Table 1 also allows us to see the problems with engagement and the variation between sites that were discussed earlier numerically.

## 4. Conclusions

The review of available evidence and reflections presented here suggest that there is a real need for interventions such as Early FDAC which reach out to pregnant mothers likely to face family court proceedings. While services for non-pregnant mothers have proliferated in the last five years, there are still relatively few services for pregnant mothers and their partners [36]. A full evaluation of the Early FDAC model, notwithstanding the challenges of recruitment and engagement outlined here, would offer valuable insight into how such services might help to deliver better outcomes for parents and children.

Building a robust recruitment pathway at the early stage of a pregnancy is essential. Encouragingly, there is some indication that new research initiatives such as 'Born into Care' [37] are beginning to translate into local authority practice through a greater level of intervention in the second trimester [38]. Winning the trust of pregnant mothers who have had previous children removed requires sensitivity and patience. This may be achieved through, for example, the greater use of 'parent mentors' who can advise parents about their own experience of recovery, and improved access to independent legal advice through a network of volunteer family lawyers.

The FDAC and Early FDAC journeys to date are a testament to the rewards and challenges of developing innovative approaches within the family justice system. As this paper has shown, it is as important to share the challenges and complexities involved in such work as to share favourable outcomes.

**Funding:** This research received no external funding.

**Institutional Review Board Statement:** Not relevant.

**Informed Consent Statement:** Not relevant.

**Data Availability Statement:** All data is included in the published article.

**Conflicts of Interest:** No conflict of interests.

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
