# Peer review of "A Proof-of-Concept Pilot for an Intervention with Pregnant Mothers Who Have Had Children Removed by the State: The ‘Early Family Drug and Alcohol Court Model’"

_societies, doi:10.3390/soc11010008_

Round 1

Reviewer 1 Report

This paper presents an interesting pilot project but would benefit from editing, and the use of more formal language. Statements should be supported by references. There are other references that could have been included. There is inadequate information provided for an evaluation. 

Author Response

Thank you for your helpful feedback

I have

  1. Provided a lot more detailed evidence for the FDAC model
  2. Made the language more formal and the tone more critical
  3. Given references to support my conclusions and abandoned some of the  conclusions that I could not support with references
  4. Presented all the reliable information and explained why more is not available

Reviewer 2 Report

Comments

Title: A proof of concept pilot for an intervention with 2 pregnant mothers who have had children removed by 3 the state: the ‘Early Family Drug and Alcohol Court 4 (FDAC) model’.

This paper describes a ‘proof of concept’ pilot of the ‘Early FDAC model’. The evaluated 6 Family Drug and Alcohol Court (FDAC) model, on which Early FDAC is based, is described in some 7 detail. Overall it is an interesting proof of concept paper on a pilot/developmental model used for helping reunite families separated by the state.

Introduction/ The FDAC Model

A detailed overview and description of the ‘Early FDAC (Family Drug and Alcohol Court) pilot model is provided. This is a ‘proof of concept’ paper which reviewed the success of the Early FDAC Model in chosen locations across England. The historical basis of the model comes from a similar model in the US.

Numerous issues are raised in the delivery of their Early FDAC Model. Are there any other similar models being used across Europe which could be referenced for comparison? Evaluating other, similar models that are being delivered in Europe may help address some of the current problems raised.

Recruitment figures seemed quite low so the overall findings would need to be generalised until more data is collected and analysed.

Also, there are other minor issues which need reviewed again:

Line 147 – spelling error: joining should be “joined”.

Section 3.1.1. Evaluation

 There is a description of a pilot study being undertaken – but that there were difficulties regarding recruitment and large numbers of drop outs. Are there any specific figures from the pilot to contextualise this?

Outcomes

Due to the limited data presented as a result of a key researcher leaving as well as funding issues, it is difficult to agree with the overall conclusion that the Early FDAC Model would be more effective than the Standard Model especially in the long-term. More reliable results on the outcomes and evaluation of this model are required in future in order to determine its long-term effectiveness.

Conclusions

While many of the study’s reliability issues are outlined here, perhaps all of the issues identified in the design and delivery of this pilot could be a suggestion for future research? The issues identified could be used to help develop the next policy and strategy at ministerial level to help improve this model process. Again, reference to other countries models from which comparisons could be drawn may be useful to include here.

Perhaps consider: Pregnancy, childcare and the family: key issues for Europe’s response to drugs

EMCDDA, Lisbon, October 2012: https://www.emcdda.europa.eu/publications/selected-issues/children_bg

Author Response

Thank you for your very helpful feedback

I have revised the manuscript to address your concerns;

  1. I have referred to the European literature on interventions with substance misusing pregnant mother (thank you for your suggested reference) and some American research on helping substance misusing mothers to be able to keep their unborn children. Nothing quite the same but useful none the less.
  2. I very much accept that it is too soon to say we know how to reduce recurrent removals.
  3. Thank you for pointing out the spelling error.
  4. I have added the figures on recruitment and engagement much earlier in the narrative 
  5. I totally agree that the evidence for Early FDAC is in no way as strong as the evaluated FDAC model and I hope that is now clear in the revised version of the manuscript
  6. You are absolutely right that addressing the issues with recruitment and engagement are key to any further trials and or evaluations of Early FDAC

Reviewer 3 Report

This manuscript presents a proof of concept for a pilot intervention (Early FDAC) with women who are pregnant who are in danger of having their children removed by the State. Based on the FDAC model, the intervention aims to work with women before giving birth to prevent having children removed and set them up for success. In the pilot, there were several problems recruiting and engaging women into Early FDAC services. The Coventry site, however, had more referrals and more cases enrolled. The ms could be improved if there was more specific information about strategies used to identify and engage women into services. It is difficult to learn from this ms how the Early FDAC could be improved upon, exactly what services are offered and needed, what challenges were faced and how the intervention tried to overcome them. The author notes the evaluator could not look at the data--are there more data to analyze than presented here? Is it possible to incorporate it into the ms? 

Suggest removing the first person writing throughout the paper and in particular in the conclusion. The conclusion is a bit speculative and should be rooted in facts. For example, if there are those (the author does not ID who thinks this) who think it is too expensive--are there data to show it is not? 

Author Response

Thank you for your helpful feedback

I have 

  1. strengthened the account of what challenges were faced and was done to try and engage the mothers and improve the design in other ways
  2. There is more data but for the most significant site it would require going back through the case notes which given the logistical challenges in the Covid crisis was not possible. 
  3. Unfortunately, there is no more available data to include in this paper at this stage
  4. I have changed the manuscript to introduce a more critical tone, removed the first person voice and dropped any speculative conclusion

Reviewer 4 Report

The article introduces a new program, the FDAC, which creates a unique court designed to regularly and functionally help at-risk mothers via the formal judicial system. The article has some promise, but as it stands, it does not add anything to the academic debate. It is descriptive, not explanatory. Below, I offer a solution to that problem.

The bottom line of the article is that the FDAC seems to be working. That much seems fairly clear. The problem, though, is the that author does not add anything to that conclusion. In fact, about one quarter of the paper is rehashing what Harwin has already said. It feels like about 25% of the article was simply copied & pasted from what someone else has already researched. That is not very original. The original data comes in section 3.1.5. There are 14 lines of original data. That is not very much.

I would expand on the original data. What KIND of families were more likely to succeed in FDAC. Is there anything about their make-up? Is there any causal factor? If that cannot be demonstrated, then the authors need to at least lay out the hypotheses and make some kind of prediction about the constellation of their respective influences. As I see it, FDAC offers a number of advantages:

1) Mothers are repeat players with the court system, specifically the same judge. Perhaps this builds trust? Perhaps it builds accountability?

2) The judicial set-up allows for an approach consistent with bureaucratic legalism, rather than adversarial legalism. (See Robert Kagan's work) Essentially, this means that the judge (rather than the litigants/lawyers) drive the process. The judge collects the facts, questions witnesses directly, etc. England is one of the few countries that has a strict adversarial process; the FDAC shows that perhaps bureaucratic legalism can work even in an environment of a deep belief in personal legal rights. (I think this, really, should be the take-home academic message of the article)

3) There are more resources available through the FDAC. To wit, whereas regular courts and social workers simply say "go to rehab," in the FDAC, there is mentorship and a team of experts all willing to help mothers. Of course FDAC litigants, then, are more likely to succeed.

4) The more informal structure allows lawyers to be taken out of the equation but for them still to be present should the mothers want to fall back on a more adversarial structure. Perhaps removing lawyers helps?

In sum, I can locate a number of reasons why FDAC is more likely to work better than regular courts. There is familiarity, more care, a state official trained to do only this, more resources available, and a structure designed to make the program work. Wouldn't it be a surprise, then, if it DID NOT work? It's no surprise that it did. So when the authors say as much, it's not all that interesting.

The interest, then, must come from the authors hypothesizing about WHY it worked. I offer 4 possible reasons. But there could be others. And in all likelihood, it is a combination of reasons. I recommend the authors lay out the advantages, rank them, and describe their individual and collective effects. I recommend summarizing the Evaluation section more succinctly (otherwise, this is not original; it's just re-telling someone else's research).

In sum, the paper needs more analysis and less reporting. More explanation and less description.

One final thing: Figure 1 should be a table with numbers instead of a figure with graphics.

I think there is a publishable paper here. I would recommend revising it in a way so that it offers more original research and/or insights.

Author Response

Thank you so much for your thoughtful criticisms 

I am confident that an article along the lines that you have suggested would be an extremely worthwhile endeavour. 

In preparing this paper my aim has been a much more modest; to describe the Early FDAC proof of concept pilot.

The Early FDAC model is still at a very preliminary phase and there is very little solid data to present, much less explain. 

I accept that the article leans heavily on published research about the 'evaluated version of FDAC'.

My understanding is that the editors thought this would help ground the preliminary Early FDAC model by linking it to the more substantially evidenced version.

Round 2

Reviewer 1 Report

Greater analysis of the literature would improve the paper further.

Author Response

With the help of the editors I have made improvements to the flow and critical tone of the paper

Reviewer 3 Report

The author's revisions have improved the ms.

Author Response

With the help of the editors I have improved the flow and critical tone of the paper and added a reference to intensive treatment

Reviewer 4 Report

I tried to give ways to frame the question theoretically--perhaps saying that a bureaucratic legalism framework can work even in places where the culture might be hostile to such a system. Or, the author could have speculated on WHY the FDAC worked...which specific items were relatively more important...and then left it to future researchers to investigate.

The author did nothing to address my concerns. That's fine--my feelings are not hurt; authors have to take papers in the direction that THEY see fit, and not how REVIEWERS see fit. But my concerns were aimed at making the paper publishable. As it stands, the manuscript offers no real new insights. It shows:

1) Other people have written some stuff. It is literally copy/pasted.

2) A program that had money and resources and was supposed to produce Outcome X produced Outcome X. This is like saying, "When the government sets up and funds a program that is specifically designed to produce Outcome X, then that outcome is more likely than when the government does not set up and fund a program specifically for that outcome." That's not very interesting.

In the end, this paper's conclusion is: "When money, time, and energy are devoted to something, usually that is better than when money, time, and energy are not devoted to something." I can't recommend publication.